# The Effectiveness of Intervening on Social Isolation to Reduce Mortality during Heat Waves in Aged Population: A Retrospective Ecological Study

**DOI:** 10.3390/ijerph182111587

**Published:** 2021-11-04

**Authors:** Stefano Orlando, Claudia Mosconi, Carolina De Santo, Leonardo Emberti Gialloreti, Maria Chiara Inzerilli, Olga Madaro, Sandro Mancinelli, Fausto Ciccacci, Maria Cristina Marazzi, Leonardo Palombi, Giuseppe Liotta

**Affiliations:** 1Department of Biomedicine and Prevention, University of Rome Tor Vergata, 00133 Rome, Italy; mosconi.claudia85@gmail.com (C.M.); desantocarolina@gmail.com (C.D.S.); leonardo.emberti.gialloreti@uniroma2.it (L.E.G.); leonardo.palombi@uniroma2.it (L.P.); giuseppe.liotta@uniroma2.it (G.L.); 2Community of Sant’Egidio, “Long Live the Elderly” Program, 00153 Rome, Italy; chiarainzerilli@gmail.com (M.C.I.); olga.madaro@gmail.com (O.M.); 3Medicine and Surgery Program, Unicamillus, Saint Camillus International University of Health Sciences, 00131 Rome, Italy; sandro.mancinelli@uniroma2.it (S.M.); fausto.ciccacci@unicamillus.org (F.C.); 4Department of Human Sciences, Libera Università degli Studi Maria Ss Assunta di Roma, 00193 Rome, Italy; marazzi@lumsa.it

**Keywords:** extreme weather, heat waves, environment and public health, aged, older adults, social behavior, interpersonal relation, social isolation, mortality, loneliness

## Abstract

Background: Heat waves are correlated with increased mortality in the aged population. Social isolation is known as a vulnerability factor. This study aims at evaluating the correlation between an intervention to reduce social isolation and the increase in mortality in the population over 80 during heat waves. Methods: This study adopted a retrospective ecologic design. We compared the excess mortality rate (EMR) in the over-80 population during heat waves in urban areas of Rome (Italy) where a program to reduce social isolation was implemented, to others where it was not implemented. We measured the mortality of the summer periods from 2015 to 2019 compared with 2014 (a year without heat waves). Winter mortality, cadastral income, and the proportion of people over 90 were included in the multivariate Poisson regression. Results: The EMR in the intervention and controls was 2.70% and 3.81%, respectively. The rate ratio was 0.70 (c.i. 0.54–0.92, *p*-value 0.01). The incidence rate ratio (IRR) of the interventions, with respect to the controls, was 0.76 (c.i. 0.59–0.98). After adjusting for other variables, the IRR was 0.44 (c.i. 0.32–0.60). Conclusions: Reducing social isolation could limit the impact of heat waves on the mortality of the elderly population.

## 1. Introduction

Extensive scientific literature has shown that climate change, particularly heat waves, induces severe health effects, and their impact on human mortality is an essential public health subject [1]. In recent decades, several heat waves have had a significant impact on health, such as the 2003 heat wave which caused approximately 70,000 deaths in Europe, and they are likely to increase in number, duration, and frequency [2]. More recently, Russia also experienced heat waves in 2010 [3] and China from 2011 to 2014 [4]. Several epidemiologic studies investigated different subgroups and various heat wave-related vulnerability factors, including individual and contextual factors. The individual factors are age, sex, or socioeconomic factors such as education, ethnicity, income or social isolation [1,5,6,7]; the environmental factors include urban design, neighborhood, and material conditions such as the availability of air conditioning [1]. These studies have seen the most significant effect, in terms of mortality, the morbidity of heat waves, and elevated temperature, on older adults, one of the most vulnerable groups.

European countries have designed public health interventions to mitigate the impact of heat waves on this sub-population, such as contingency and cooling plans, urban interventions, and adaptation of buildings and cities to reduce heat stress and exposure [1]. An example is the “European WHO heat health action plan” that includes a warning system during heat waves, plans for emergency measures, actions aimed at reducing high environmental temperatures, as well as greening activities [1]. In Italy, the Ministry of Health in 2005 launched the National Plan for the prevention of the effects of heat on health through projects from the National Center for Disease Prevention and Control (Ccm) and with the coordination of the National Competence Center Department of Epidemiology SSR Lazio Region (DEP Lazio). The aim was to provide operational guidelines for the prevention of the effects of heat waves on health. The plan is divided into various levels of alarm and surveillance. In the summer of 2021, an activity plan was established in relation to the COVID-19 pandemic [8]. During a heat wave in 2014, Spain introduced a “national plan for preventive actions against the effects of excess temperatures on health” that provides weather forecasts, preventive information to the general population and specific high-risk groups, and activation of emergency services [9].

These programs target environmental aspects and living contexts but are not explicitly aimed at the most vulnerable groups, such as the elderly. Social isolation directly impacts the health of the elderly [10,11,12,13,14] and increases the risk of death during heat waves [15]; however, only a few interventions have been designed to address this issue, and there is no clear evidence of their effectiveness [10,16]. With a more targeted strategy, it is possible to reduce mortality and morbidity in this sub-population by addressing modifiable factors that increase vulnerability, such as social isolation.

We hypothesize that if we reduce social isolation, considered as a mediating factor between the heat waves and mortality, the increase in mortality during heat waves will be consequently reduced. Therefore, this study aims to evaluate the effectiveness of an intervention to reduce social isolation in the over-80 population in order to reduce the mortality in this population during heat waves.

In the literature, to date, there are no interventions that try to reduce the impact of heat waves by intervening on social isolation [10].

## 2. Methods

This study focuses on the impact of a program to reduce social isolation during heat waves between 2014 and 2019 in Rome, Italy. The study takes place in the first municipality of Rome, the smallest in the city and with a slight variation in the wealth distribution among the areas of the municipality [17]. The area is divided into eight administrative urban zones, one of which is not inhabited because it is an archaeological area. In 2004, a not-for-profit organization, the “Community of Sant’Egidio”, started a program in three of these urban zones (Trastevere, Testaccio and Esquilino) aimed at limiting mortality due to heat waves by reducing social isolation among the population aged more than 80 called “Long Live Elderly!” (LLE). In three urban zones (Centro Storico, Aventino and XX Settembre) the intervention was not carried out, and in the Celio area LLE started in 2016.

### 2.1. The Intervention

LLE is a program designed to support elderly populations during heat waves by trying to prevent and manage the harmful effects of heat waves on health. The program aimed to contact all older adults living in the chosen urban zone (universalistic approach) and detect the ones who were socially isolated or sick to offer them a periodic assessment of their socio-health needs, inform them about health promotion campaigns and behaviors to be adopted during heat waves, and provide them with assistance in managing their daily tasks. LLE aimed to strengthen the community network around the target population with voluntary actions and increased community awareness around the needs of the elderly with a proactive approach. Phone calls were made with a maximum frequency of once every two weeks to those who joined the initiative with specific, informed consent. Based on the risk of an adverse event, an individual care plan was drawn up. This plan also included services that the program could not provide directly but would facilitate in their provision to patients. The activities foreseen by the LLE program increased during the heat waves. The target population was reached by telephone and, if necessary, the staff carried out a home visit to meet specific needs (e.g., deliver food and/or medicines). For further details on the elements of the intervention program, please refer to a previous publication on this intervention [18].

In 2014, the municipality of Rome began to collect data on mortality for the resident population disaggregated by age and small urban areas (urban zones). Therefore, it is possible to compare mortality in the over-80 population in the areas where the LLE program took place with the mortality in the areas where it was not operative. Data on mortality were provided by the statistical office of the municipality of Rome.

At the outset, the program proposed the intervention on the population over 75 years old. Starting from 2016, only the population over 80 years old that identified as more vulnerable to the effects of heat waves was selected as the target population of the proposed intervention.

This study adopted a multi-group retrospective ecological design. We compared aggregated mortality in the urban zones where the LLE program took place with similar areas where the intervention did not occur and evaluated aggregated indicators related to the areas and not individuals.

### 2.2. Population

The general population of the study refers to adults over 80 living in an urban context. The sample included 7 urban zones in the first municipality of Rome and considered aggregated data related to the over-80 population. Zones have been included in the intervention group if the LLE program was present and in the control group if they belonged to the same municipality but the program was absent. The attribution of zones to the intervention or control group is not randomized: the Community of Sant Egidio initiated the LLE program to counteract loneliness in the Trastevere zone, where the organization’s headquarters are located. The program was then replicated in the other urban zones of the same municipality, following a criterion of feasibility and a need assessment. The areas included in the two groups are represented in Figure 1, reporting the map of the urban zones composing the Rome I municipality.

### 2.3. Measures or Variables

The primary outcome of the analysis is the excess mortality in each summer period (from June to September) from 2015 to 2019 compared to the year 2014 in which there were no heat waves.

Other variables associated with outcome are the dose of the exposure, measured by the number of days of heat waves during the summer period, and other socio-economic characteristics of the urban areas’ populations. The criteria for defining a heat wave are those established by the Italian civil protection agency: very high temperatures occurring for several consecutive days, high humidity, solar radiation, and a lack of ventilation. A heat wave is defined concerning the climatic conditions of a specific city, so there is no threshold temperature valid at all latitudes. The national forecasting and warning system, managed by the Ministry of Health with the technical-scientific contribution of the National Competence Center (CCN), Department of Epidemiology SSR Lazio Region, guarantees the monitoring of meteorological conditions associated with a health risk. The specific city forecast and alarm systems (also called Heat health watch warning systems—HHWWs) can evaluate the impact of temperature on health through the integrated analysis of climatic conditions, historical data of mortality, and meteorological variables. The results of the forecast and alarm systems are summarized in a specific city’s daily bulletin that reports adverse health conditions for the same day and the following two days, through four graduated risk levels defined by the severity of the scheduled events from zero to three. Level three is considered a heat wave and it is defined by high-risk conditions that persist for three or more consecutive days [18,19].

In this study, we observed the number of days when a bulletin indicated a level three risk (heat wave). We also considered each summer period as an a high-level of exposure if the number of days of heat waves were higher than the median value of the period from 2015 to 2019 and as a low-level of exposure if they were below the median value.

Other variables related to the urban zones that can be associated with excess mortality in the event of a heat wave are the socio-economic situation of the urban area, winter mortality, and the percentage of the population over 90. The first is a complex concept and can be measured by various statistical indicators. In this study, the average cadastral income per 100 square meters in the study period was used as a proxy for the wealth level of the urban area, as in similar studies [1,9,18]. The second variable—winter mortality—can influence summer mortality according to what is called the “harvesting” effect: when mortality is higher in the winter period, it can be lower in summer because most frail individuals have already died [2,6,7,18]. Winter mortality was calculated as the rate of deaths in the population > 80 divided by the population > 80 in the months from November to February. The third variable (proportion of people over 90) could affect mortality since the probability of dying increases with age.

### 2.4. Statistical Analyses

Statistical analysis was performed using software “R” version 4.1.0 [20] (packages epitools [21], pubh [22] and sjPlot [23] to report results of analysis).

The unit of the analysis was the summer periods per year per urban zone. The dependent variable is a rate given by the difference between the number of deaths in the summer period and the number of deaths in the summer of 2014 divided by the average number of citizens over 80 in the period from 2015 to 2019 as expressed in the following formula.
(1)$$\mathrm{Primary}\;\mathrm{outcome}\;\left(\mathrm{per}\;\mathrm{years}\;2015–2019\right)=\frac{\mathrm{Deaths}\;\mathrm{number}\;\mathrm{in}\;\mathrm{summer}\;\mathrm{per}\;\mathrm{year}-\mathrm{Number}\;\mathrm{of}\;\mathrm{deaths}\;\mathrm{in}\;2014\;\mathrm{summer}}{\mathrm{Average}\;\mathrm{number}\;\mathrm{of}\;\mathrm{over}\;80\;\mathrm{individuals}}$$

We used a two-sample test of proportions to assess the difference between groups. For the other continuous variables—the number of days with heat waves per season, cadastral income, winter mortality, and population over 90—the mean was reported and the difference between the two groups was tested with Student’s *t*-test for normally distributed variables and the Wilcoxon Rank-sum test for non-normally distributed variables. To evaluate the association of the various variables with the dependent variable, we performed both a univariate and multivariate Poisson regression including all the variables. Poisson regression defines a generalized linear model for regression analysis when the dependent variables are count data, rates, or contingency tables [24]. We calculated the incidence rate ratio (IRR), the confidence interval for a significance level greater than 95%, and the value of the Nagelkerke R-square to estimate the goodness-of-fit (GOF) of the model.

In accordance with the University of Tor Vergata guidelines, ethical clearance was not required given the study employed pre-existing public aggregated data and did not involve human participants.

## 3. Results

The summer periods considered are for 6 years for seven districts for a total of 42 observations. Twenty-two observations fall within the intervention group (18 observations in the zones where the program was implemented plus four observations in the Celio zone from 2016 to 2019) and 20 observations fall within the control group (18 observations in the zones where the program was not carried out plus two in the Celio zone from 2014 to 2015). The descriptive statistics of the two groups are shown in Table 1.

The main outcome, i.e., the excess mortality rate compared to 2014 in which there were no heat waves, is represented in Figure 2. For urban zones included in the intervention group, the excess mortality rate was lower than the one registered in the control group for the years observed.

In 2014, there were 96 and 64 deaths, respectively, in the over-80 population in the intervention and control neighborhoods. In the following 5 years, the excess deaths were 102 (intervention) and 119 (controls) with a rate of 2.70% vs. 3.81%. The rate ratio is 0.70 (c.i. 0.54–0.92, *p*-value 0.01).

The results of the univariate Poisson regression for the group (intervention vs. control), level of heat waves (high-level vs. low-level), winter mortality, and cadastral income per 100 m^2^ are shown in Table 2. All variables are associated with the excess mortality rate (except for age > 90), and the variables group and the level of heat wave are those with the greatest effect (IRR 0.76 and 1.49, respectively).

The results of the multivariable regression are shown in Table 3. Being a part of the urban zones with intervention and a higher cadastral income is associated with a lower risk of mortality (IRR 0.44 and 0.77, respectively), while a high level of heat waves (number of days greater than the median value) is associated with a greater excess of mortality.

## 4. Discussion

The most important result is that, although there was an increase in mortality, in the urban areas where intervention was implemented to reduce the isolation of the elderly, during the summer heat waves, this was not as marked as in the areas without intervention. This result occurred for each of the 5 years considered in this study. The association between the presence of intervention and lower excess mortality was even more significant after adjusting for the level of heat waves, the cadastral income of the neighborhood, and the winter mortality (IRR 0.44, CI 0.32–0.60, *p*-value < 0.001).

Our findings are in line with results from previous studies that have analyzed heat wave-related mortality [2,5,9]. In several epidemiological studies on heat-related mortality, various subgroups have been identified as the most severely affected and defined as vulnerable [1,6,10]. Social isolation is included among the vulnerability factors [1]. Our study differs from previous studies because, in the literature, there are no data on social isolation correlation with mortality during heat waves. Our results confirm that effective action to reduce social isolation could have a considerable impact on mortality.

There are still doubts whether interventions aimed at counteracting social isolation are really achieving their aims, [8] and even more about their effectiveness in improving health outcomes among the beneficiaries. This study supports the effectiveness of increasing social connectedness at both population and individual levels by targeted interventions that seem to fill the gap created by social isolation, which is a well-known mortality risk factor, and reduce negative health outcomes in the individuals involved in the program.

Based on our findings, research and other prevention programs aimed at reducing social isolation are recommended. Conducting programs to reduce social isolation can have a significant impact on the health of the elderly; moreover, it is advisable to investigate the effectiveness and cost-effectiveness of the interventions that can be implemented in this context and others. Various urban interventions have been described in the literature to increase urban vegetation, or change the urban structure of cities [1,2]. However, we assume that interventions aimed at improving social isolation could be cheaper, easier to implement, and reach the most vulnerable population more directly.

The main limitation of this study is related to the observational and ecological design. There is the possibility of an ecologic bias, that is, to attribute to individuals both the exposure and the outcome that are instead linked to a group. A significant limitation is that heat waves do not equally affect all the elderly in the same urban zone. For example, even if the areas are similar and bordering, it is possible that the exposure—the increase in temperature—is perceived differently depending on the type of houses, the presence of vegetation that can vary from one street to another, or the presence of air conditioners. Therefore, these are both environmental and individual confounders not included in this analysis. Another bias is that the design does not consider individual health-related confounders, such as the presence of co-morbidities in the elderly. A list of potential confounders that we did not consider due to the lack of necessary data is reported in Box 1. Regarding the selection of interventions and controls, non-randomization exposes the risk of selection bias. In this case, however, no characteristics of the urban zones have been identified that could increase the probability of being included in the intervention that are also correlated with lower mortality in the elderly. Conversely, the Community of Sant’Egidio organization has chosen to intervene first in the areas where the need assessment showed more social isolation and vulnerability in older adults. This aspect is confirmed by the fact that the excess mortality is lower in the urban areas with a higher economic status, and that after adjusting for the cadastral income, the correlation with the intervention is more significant. Finally, a retrospective study does not allow us to infer causality but only a correlation.

Box 1Box with a list of possible confounders not included in the analysis.
Possible confounders not included in the analysisPopulation level mitigation measuresRegional health systems’ capacitiesNumber of phone calls from friends or neighbors in the general populationNumber of visits from friends or neighbors in the general populationFrequency and duration of the above visits/callsAction performed by the visitors (bring food or medication, drive the partici-pants to medical appointments and/or to pharmacies to collect medications, drive respondents to go shopping)Illness distribution in the two populationsDrug consumption in the two populationsCauses of deaths in the two populationsImpact of these factors on nursing home residents included in the study


On the other hand, one of the strengths of this study is that, while adopting an ecological design, the areas considered are very small, and therefore they are more likely to be homogeneous with each other. In addition, the intervention in the selected urban zones reached all individuals over 80, not just a portion of them. Furthermore, an ecological design is suitable for the purpose of this research since heat waves do not affect single individuals but groups living in the same area; social isolation is both an individual and a collective issue. Moreover, this study is the first to analyze the effectiveness of such a program, and despite the limitations of the study, it is possible to consider the hypothesis that reducing social isolation can also reduce mortality. It is, therefore, a first step that could be followed by a prospective study able to consider more confounders and verify other principles of causality.

## 5. Conclusions

The data on mortality by age group in seven urban areas of Rome (Italy) show a correlation between a program to reduce social isolation and a lower increase in deaths during the summer periods characterized by heat waves in the over-80 population. The correlation remains confirmed after adjusting for cadastral income, winter mortality, and the percentage of citizens over 90 in the two groups. Prospective randomized studies are needed to verify the causal link between reducing social isolation and lower excess mortality during a heat wave. If this link is confirmed, intervening in social isolation can be a highly effective and low-cost intervention, compared to other urban interventions, to reduce the impact of heat waves on the elderly population.

## Figures and Tables

**Figure 1 ijerph-18-11587-f001:**
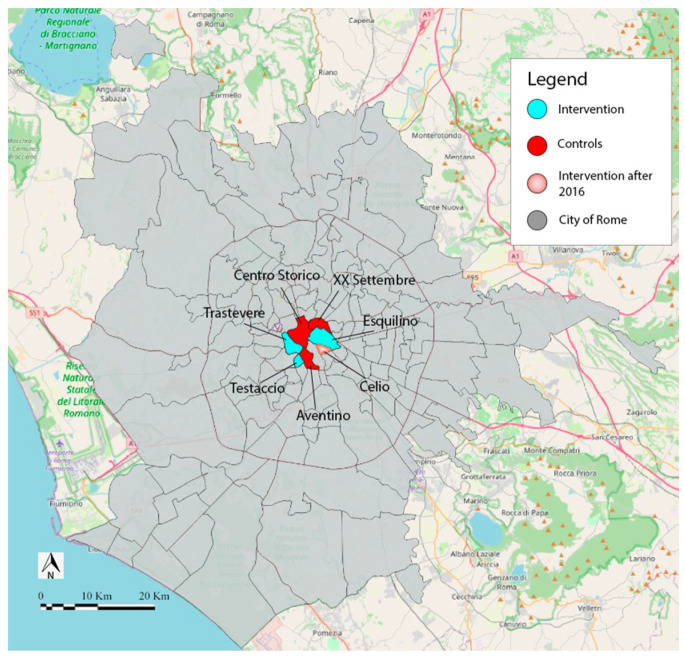
Map of the urban zones of the first municipality of Rome, Italy.

**Figure 2 ijerph-18-11587-f002:**
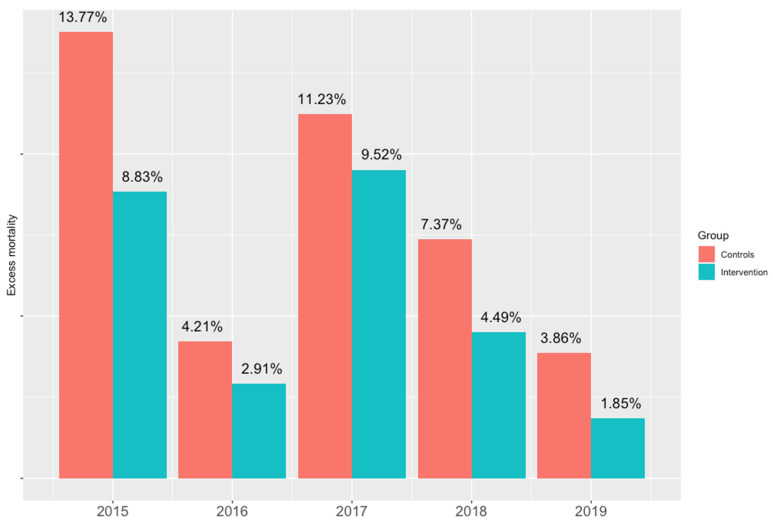
Excess mortality during the summer period with respect to 2014 per year (intervention vs. controls).

**Table 1 ijerph-18-11587-t001:** Descriptive statistics of the urban zones.

Independent Variables	Controls	Intervention	Total	*p*	Distribution
	(*n* = 20)	(*n* = 22)	(*n* = 42)		
Average population over 80 (2015–2019)	720.2 [650.5; 1479.8]	760.8 [602.4; 2146.7]	720.2 [602.4; 1479.8]	1.000	non-normal
Average age	48.1 [47.6; 49.2]	47.1 [47.0; 48.4]	48.1 [47.1; 48.4]	0.034 *	non-normal
Percentage of population over 75	13.6 [12.3; 15.2]	11.6 [10.9; 14.0]	13.3 [11.6; 14.0]	0.006 **	non-normal
Percentage of population over 90	11.2 [9.6; 11.7]	8.8 [7.4; 11.4]	9.6 [8.8; 11.4]	0.001	non-normal
Ageing index × 100 (population > 65/population < 15)	255.2 [240.6; 273.9]	228.2 [219.0; 245.8]	240.6 [228.2; 255.2]	0.000 ***	non-normal
Dependency index × 100 (population > 65 + population < 15 over total)	55.7 [53.7; 56.6]	52.6 [52.1; 57.3]	55.7 [52.6; 57.3]	0.366	non-normal
Masculinity index (man/women × 100)	94.5 [87.3; 100.0]	94.5 [83.0; 97.3]	94.5 [84.8; 97.3]	0.093	non-normal
Cadastral income (100 sq-meters)	17.9 [16.4; 19.8]	16.7 [13.7; 17.3]	16.7 [16.4; 17.9]	0.015 *	non-normal
Winter mortality	34.0 ± 10.9	36.4 ± 8.2	35.3 ± 9.5	0.464	normal

* *p*-value < 0.05, ** *p*-value < 0.01, *** *p*-value < 0.001.

**Table 2 ijerph-18-11587-t002:** Univariate Poisson regression. Dependent variable: excess mortality rate.

Variables	IRR	IRR	IRR	IRR	IRR
(Intercept)	0.01 *** (0.01–0.01)	0.01 *** (0.00–0.01)	0.00 *** (0.00–0.01)	0.10 *** (0.02–0.38)	0.02 *** (0.01–0.04)
group: Intervention	0.76 * (0.59–0.98)				
High heat wave level		1.49 ** (1.11–2.03)			
Winter mortality			1.02 * (1.00–1.04)		
Cadastral income (100 sq-meters)				0.87 *** (0.80–0.94)	
Proportion of people over 90					0.94 (0.85–1.03)
Observations	30	30	30	30	30
R^2^ Nagelkerke	0.141	0.220	0.165	0.325	0.053

* *p* < 0.05 ** *p* < 0.01 *** *p* < 0.001.

**Table 3 ijerph-18-11587-t003:** Multivariable Poisson regression.

Variables	Excess Mortality Rate	
	Incidence Rate Ratios	CI
(Intercept)	1.82	0.21–14.55
group: Intervention	0.44 ***	0.32–0.60
High heat wave level	1.57 **	1.16–2.17
Cadestral income	0.77 ***	0.70–0.85
Winter mortality	1	0.98–1.02
Proportion of people over 90	0.92	0.84–1.01
Observations	30	
R^2^ Nagelkerke	0.802	

** *p* < 0.01 *** *p* < 0.001.

## Data Availability

The database generated for this study is available at OSF (www.osf.io) (accessed on 3 November 2021) with doi:10.17605/OSF.IO/8MGXA.

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
