# Peer review of "The Effectiveness of Intervening on Social Isolation to Reduce Mortality during Heat Waves in Aged Population: A Retrospective Ecological Study"

_ijerph, 2021, doi:10.3390/ijerph182111587_

Round 1

Reviewer 1 Report

This is an interesting study that allows evaluating a social support intervention for the elderly to avoid excess mortality caused by heat waves, observed in recent years in Europe. Ecodesign has its limitations, which the authors describe perfectly. There are some details of the method that are necessary to specify better:
For the year 2014 the denominator of the rate should be the population over 80 of this same year, not the average 2015-2019. This is not described properly in the text.

In the table 1: It should be described in detail which population and years are refering with "average population per urban zone" , on the table or as a foot note. Total population? Oven 80? Which years?

The first paragraph of the discussion section can be improved as it leads to confusion in the way it is written.

I suggest:
The most important results is that in the urban areas where an intervention was implemented to reduce the isolation of the elderly, although there was an increase in mortality, during the summer heat waves, this was not as marked as in areas without intervention.

Please check the comments in the file.

Author Response

Thank you for your comments/suggestion. Please find here the responses point-by-point

  • For the year 2014 the denominator of the rate should be the population over 80 of this same year, not the average 2015-2019. This is not described properly in the text.

We did not calculate the rate in 2014. The year 2014 was considered as a base year because, in that year, there were no heat waves (exposure). Therefore, for 2015 - 2019, we calculated the number of deaths in excess over those in 2014 for each urban area. To obtain the ratio, we divided this number by the average population in that period. A different approach could have been to use as a denominator the population in each year (2015-2019), but the result would have been similar since there were no significant changes in the population (denominator). To make this methodology clearer, we have corrected the text of the first paragraph in the "Measures or Variables" section and added a formula immediately after the paragraph.

  • In the table 1: It should be described in detail which population and years are refering with "average population per urban zone" , on the table or as a foot note. Total population? Oven 80? Which years?

Thanks for this suggestion. It was an oversight. We have improved the table.

  • The first paragraph of the discussion section can be improved as it leads to confusion in the way it is written. 

I suggest:

The most important result is that in the urban areas where intervention was implemented to reduce the isolation of the elderly, although there was an increase in mortality, during the summer heat waves, this was not as marked as in areas without intervention.

Thank you, we accepted your suggestion

Even if the system does not allow to upload a new version of the manuscript before the acceptance of replied from the reviewers, you could find useful to verify the improvements in the attached draft.

Thank you

Reviewer 2 Report

The impact of heatwaves in the aged population is an important issue in public health, global warming. This study aims to evaluate the effectiveness of a community intervention for elderly residents in some urban areas of Rome. The results showed that the excess mortality rate among the elderly population in the intervention areas was decreased compared with the non-interventional area.

  1. Although the topic is important; however, the findings are not convincing because the causality is not clear. Did the interventional program include any item to improve the elders’ knowledge, attitude, or practice to cope with the heat stress? Introduce the content of the interventional program.
  2. Authors should discuss potential confounding factors that may affect mortality between the interventional and non-interventional areas. A table comparing the demographic characteristics of the two groups may be helpful.

Author Response

Thank you for your comments/suggestion. Please find here the responses point-by-point

  • Although the topic is important; however, the findings are not convincing because the causality is not clear. 

We agree that causality is not clear. In fact, one of the limitations of observational ecological design is precisely that with this kind of studies it is not possible to define a clear causal link. We clearly indicated this limitation and interpretation in the discussion. 

However, not being possible a randomized prospective study for his costs, and being this one the first study in the literature evaluating interventions on social isolation as a mediating factor in case of heath waves also the results in terms of association/correlation could be valuable. This result, in our opinion, can increase the evidence supporting interventions to reduce social isolation in the elderly, but can also be a good base to carry out other studies with a more rigorous design that allow detecting causality.

  • Did the interventional program include any item to improve the elders’ knowledge, attitude, or practice to cope with the heat stress? Introduce the content of the interventional program.

Thanks for your suggestion, we have added a further clarification on the program's functioning in paragraph 2 of the methods section. Furthermore, the text refers to the publication "Social Interventions to Prevent Heat-Related Mortality in the Older Adult in Rome, Italy: A Quasi-Experimental Study (Giuseppe Liotta et al.)". The program is explained in detail. We thought it helpful not to go into too much detail in this paper but to refer to the reading of the previous publication.

  • Authors should discuss potential confounding factors that may affect mortality between the interventional and non-interventional areas. 

Thanks for the suggestion. We have specified the confounding factors and we also added a box with a list of potential confounding factors not included in the analysis because related data were not available

  • A table comparing the demographic characteristics of the two groups may be helpful.

Thank you for the suggestion. We added some demographic data of the two groups in table 1 with the descriptive statistics

Even if the system does not allow to upload a new version of the manuscript before the acceptance of replied from the reviewers, you could find useful to verify the improvements in the attached draft.

Reviewer 3 Report

The authors present a sound and relevant study and should be commended by their work.

The text is well written and it was very interesting to read.  It is my opinion that the text can be published with minor changes. Despite these minor comments I leave some suggestions that may help to elevate the overall quality of the text, if the authors opt to apply them.

Comments:

In the introduction the authors refer to the heatwave of 2003, being one of the most studied heatwaves it was expected, but more recent heatwaves could have been mentioned.

The map does not has scale or north arrow

The method section misses reorganization. I recommend splitting it in subtopics.

The independent variables need to be better explained (how are they calculated)

The dependent variable is not properly written on the first paragraph of page 4 (it is after though)

The statistical model needs to be more detailed (e.g. is it a linear or a nonlinear model?).

In the discussion, the first result is not that heatwaves are associated with increased mortality. I understand that the authors can derive this information from the methods applied, but they are not adequate for this inference. I would suggest them to focus on the main result: the impact of the intervention

When they compare the results I would suggest doing it with studies from Italy or from other Mediterranean countries.

The introduction of Social capital is not properly framed, in my opinion

Further suggestion:

This study is mainly about the impact of an intervention through a comparison. I suggest testing models that allow random intercepts and random slopes i.e the control group and the intervention group do not start at the same level and do not evolved at the same level

Author Response

Thank you for your comments/suggestion. Please find here the responses point-by-point.

  • In the introduction the authors refer to the heatwave of 2003, being one of the most studied heatwaves it was expected, but more recent heatwaves could have been mentioned.

Thank you for your suggestion. In the introduction, we also mentioned the critical heat wave that affected Russia in 2010 and China in 2011-2014.

  • The map does not has scale or north arrow

Thank you for the suggestion. We added the graphics elements to the map

  • The method section misses reorganization. I recommend splitting it in subtopics.

Thanks for the advice. Done.

  • The independent variables need to be better explained (how are they calculated)

We added more details on how winter mortality is calculated. The average cadastral income per 100 square meters and proportion of individuals over 90 per urban areas were not calculated from the authors but data was provided by the statistical office of Rome. 

  • The dependent variable is not properly written on the first paragraph of page 4 (it is after though)

We added more details on how we calculated the dependent variable as required by another reviewer in the section on statistical analysis. We did not report also in the paragraph on page 4 to avoid redundancy. 

  • The statistical model needs to be more detailed (e.g. is it a linear or a nonlinear model?).

We used a Poisson Regression: a generalized linear model form of regression analysis used to model count data and contingency tables. We added an explanation with a reference in the text. 

  • In the discussion, the first result is not that heatwaves are associated with increased mortality. I understand that the authors can derive this information from the methods applied, but they are not adequate for this inference. I would suggest them to focus on the main result: the impact of the intervention

Well noted. We modified the discussion accepting this suggestion and others from other reviewers

  • When they compare the results I would suggest doing it with studies from Italy or from other Mediterranean countries.

We added some references to the previous studies in these geographical areas. Actually, no studies have been performed on the role of reducing social isolation to reduce the effect of heat waves 

  • The introduction of Social capital is not properly framed, in my opinion

That’s right. The use of the locution “social capital” was inappropriate or not clear without a better definition in this context. I meant an antonym of “social isolation”. I think the best word is “social connectedness” so I replaced “social capital” with that. 

Further suggestion:

This study is mainly about the impact of an intervention through a comparison. I suggest testing models that allow random intercepts and random slopes i.e the control group and the intervention group do not start at the same level and do not evolved at the same level

Thank you for the suggestion. In fact, we initially thought of using an interrupted time series analysis that is indicated to evaluate if there are different trends over time. However, this project was not feasible because the municipality of Rome started calculating the disaggregated mortality for urban areas only from 2014, so it was not possible for us to build a sufficiently populated time series of data on the period before the start of the intervention. For this reason, we have adopted Poisson regression as a tool which, however, does not allow us to estimate slopes or intercepts but only differences between count data and rates.

However, in the future, we plan to seek the data necessary to carry out an analysis of this kind, or a prospective randomized study that allows us to reach more robust and generalizable conclusions.

_____

Even if the system does not allow to upload a new version of the manuscript before the acceptance of replies from the reviewers, you could find it useful to verify the improvements in the attached draft.

Round 2

Reviewer 2 Report

  1. Typo in the Method section (XX Settember)?
  2. Table 2, 3. change "level : High" to "high heave waves level"